# Alleviate Anchor-Shift: Explore Blind Spots with Cross-View Reconstruction for Incomplete Multi-View Clustering

**Suyuan Liu**[1]    **Siwei Wang**[2]    **Ke Liang**[1]    **Junpu Zhang**[1]    **Zhibin Dong**[1]
**Tianrui Liu**[1]    **En Zhu**[1]    **Kunlun He**[3]    **Xinwang Liu**[1*]

[1]National University of Defense Technology, Changsha, China
[2]Academy of Military Sciences, Beijing, China
[3] Chinese PLA General hospital, Beijing, China
{suyuanliu, enzhu, xinwangliu}@nudt.edu.cn

## Abstract

Incomplete multi-view clustering aims to learn complete correlations among samples by leveraging complementary information across multiple views for clustering. Anchor-based methods further establish sample-level similarities for representative anchor generation, effectively addressing scalability issues in large-scale scenarios. Despite efficiency improvements, existing methods overlook the misguidance in anchors learning induced by partial missing samples, *i.e.,* the absence of samples results in shift of learned anchors, further leading to sub-optimal clustering performance. To conquer the challenges, our solution involves a cross-view reconstruction strategy that not only alleviate the anchor shift problem through a carefully designed cross-view learning process, but also reconstructs missing samples in a way that transcends the limitations imposed by convex combinations. By employing affine combinations, our method explores areas beyond the convex hull defined by anchors, thereby illuminating blind spots in the reconstruction of missing samples. Experimental results on four benchmark datasets and three large-scale datasets validate the effectiveness of our proposed method.

## 1 Introduction

In multi-view learning, data collected from different sensors or media often suffer from missing values [1, 2, 3, 4]. For example, remote sensing images collected from various sensors may experience partial missing due to channel noise. Traditional multi-view learning methods cannot directly handle these missing values [5, 6, 7]. To address this issue, incomplete multi-view learning methods have been developed to leverage the available data from all views to perform downstream tasks [8, 9, 10]. Among these, incomplete multi-view clustering (IMC) relies on view consistency and complementarity, effectively enabling the partitioning of data with missing values [11, 12, 13].

Existing IMC methods can be categorized into three types based on their approach: similarity-based, imputation-based, and matrix decomposition-based methods. Similarity-based methods recover a relationship matrix among all samples using available data in each view [14, 15, 16, 17, 18]. Imputation-based methods first fill in the missing parts, transforming the problem into a complete multi-view clustering problem [19, 20, 21]. Matrix decomposition-based methods map samples from all views into a common latent space to construct a unified representation for clustering [22, 23, 24, 24, 25]. However, these methods all require constructing an $n \times n$ similarity matrix,

---

*Corresponding Author

38th Conference on Neural Information Processing Systems (NeurIPS 2024).

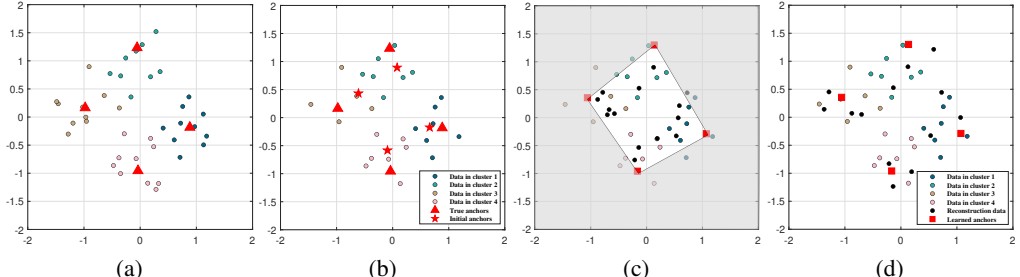

Figure 1: (a)Anchors learned in complete data. (b)Anchors initialized in incomplete data. (c)Data reconstructed with convex combination. (d)Data reconstructed with affine combination.

resulting in an $\mathcal{O}(n^2)$ space complexity, which limits their application to large-scale scenarios [26, 27, 28].

In contrast, anchor-based IMC methods reduce space complexity to $\mathcal{O}(n)$ by learning a small number of representative anchors from the data and replacing the full similarity matrix with relationships between anchors and samples [29, 30, 29, 31]. Despite addressing scalability, these methods overlook the anchor-shift problem caused by missing data. As shown in Fig. 1(a)(b), anchor learning can be misled by missing data, resulting in a discrepancy between learned anchors and those from complete data. This problem diminishes the representational capacity of anchors and causes the anchor graphs to be misaligned across views, thereby compromising the clustering performance.

In this work, we propose a novel Anchor-based Incomplete Multi-view Clustering with Cross-View Reconstruction strategy, termed AIMC-CVR. AIMC-CVR encompasses two key modules designed to resolve the anchor-shift problem in anchor-based IMC. The first module, the cross-view anchor learning module, is dedicated to mitigating the anchor-shift problem by learning a complete anchor graph. Specifically, we designed a symmetrized cross-view projection mechanism to ensure dimensional consistency across view pairs. By leveraging the relationships between anchors and samples across different view pairs, we constructed a complete anchor graph.

The second module, the affine combination-based reconstruction module, iteratively updates the anchors with available and reconstructed data. Existing anchor-based IMC methods often use convex constraints to build relationships between anchors and samples [32, 33, 34, 35], leading to blind spots in sample reconstruction. As shown in Fig. 1(c), samples reconstructed based on convex combinations are restricted to the convex hull of the anchors. Instead of relying on convex combinations, our innovative approach utilizes an affine combination-based reconstruction strategy. This strategy broadens the scope of sample reconstruction, as depicted in Fig. 1(d), allowing for a more comprehensive and accurate representation of missing data.

Our main contributions are as follows:

- By employing cross-view anchor learning and affine combination-based reconstruction, we effectively alleviate the anchor-shift problem in missing data scenarios.
- Unlike traditional sample-level imputation methods, we reconstruct missing samples with anchors and anchor graphs, significantly reducing reconstruction complexity. Affine combinations further explore blind spots in sample reconstruction, as demonstrated by theoretical analysis and experimental results.
- Comparative experiments with state-of-the-art IMC and anchor-based IMC methods validate the effectiveness and superiority of AIMC-CVR.

## 2 Related Work

**Cross-View Learning.** Cross-view learning is a specialized form of multi-view learning that leverages interactions between paired views to learn cross-view representations, enabling the exploration of finer-grained inter-view relationships [36, 37, 38]. For instance, Tang et al. ensure local structural

consistency across views by simultaneously constructing similarity graphs between pairs of views and within each view [39]. Similarly, Liu et al. leverage the fact that samples are always present in at least one view, constructing a complete cross-view similarity matrix based on relationships within and between views [40]. These methods demonstrate the potential of cross-view learning to maintain structural consistency and fully utilize the available data. In the next section, we will explore the application of cross-view learning strategies to multi-view anchor learning, focusing on view complementarity to construct a complete anchor graph and mitigate the anchor-shift problem in scenarios with missing data.

**Sample-Level Imputation.** Imputation-based multi-view clustering methods first impute the missing data before clustering [41, 42, 21]. Typically, existing methods perform sample-level imputation by leveraging multi-view information to learn complete similarity relationships between samples. For example, Yin et al. reconstruct missing data based on the decomposition matrix and sample similarity relationships [20]. Liu et al. integrate data imputation and clustering within a unified optimization framework [19]. However, these sample-level imputation methods typically rely on similarity matrices of size $\mathcal{O}(n^2)$, which limits their scalability for large datasets.

## 3 Method

In this section, we introduce the Anchor-based Incomplete Multi-view Clustering with Cross-View Reconstruction (AIMC-CVR) strategy. We begin by introducing the two core modules: the cross-view anchor learning module and the affine combination-based reconstruction module. We provide a theoretical analysis that demonstrate the merits of our proposed affine combination-based reconstruction strategy. Finally, we present the overall objective function of the algorithm and propose a four-step alternating iterative algorithm to solve the corresponding optimization problem.

### 3.1 Cross-View Anchor Learning Module

Given multi-view data $\left\{ \mathbf{X}^{(p)} \in \mathbb{R}^{d_p \times n} \right\}_{p=1}^{v}$, where $d_p$ and $n$ denotes the dimension of the data and the total number of samples, respectively.

In the context of incomplete data scenarios, the data matrix $\mathbf{X}^{(p)}$ for the $p$-th view can be partitioned into two distinct subsets, *i.e.*, $[\mathbf{X}_o^{(p)}, \mathbf{X}_m^{(p)}]$, where $\mathbf{X}_o^{(p)} \in \mathbb{R}^{d_p \times n_p}$ representing the existing portion of the data, and $\mathbf{X}_m^{(p)} \in \mathbb{R}^{d_p \times (n-n_p)}$ is the missing portion. The observed part is obtained by applying an index matrix $\mathbf{G}^{(p)} \in \{0,1\}^{n \times n_p}$ to the complete data matrix, such that $\mathbf{X}_o^{(p)} = \mathbf{X}^{(p)} \mathbf{G}^{(p)}$. The index matrix $\mathbf{G}^{(p)}$ encodes the presence of samples, where indicates that the $i$-th sample in the complete dataset corresponds to the $j$-th ranked existing sample in the observed subset $\mathbf{X}_o^{(p)}$. Based on the samples present in each view, we can learn the anchors and their corresponding anchor graphs as follows:

$$
\min_{\mathbf{A}^{(p)}, \mathbf{Z}^{(p)}} \sum_{p=1}^{v} \left\| \mathbf{X}_o^{(p)} - \mathbf{A}^{(p)} \mathbf{Z}^{(p)} \mathbf{G}^{(p)} \right\|_{\mathbf{F}}^2 ,
$$
$$
\text{s.t. } \mathbf{Z}^{(p)\top} \mathbf{1} = \mathbf{1}, \mathbf{Z}^{(p)} \geq 0.
\tag{1}
$$

The anchor graph $\mathbf{Z}^{(p)}$ is incomplete, representing only the relationship between existing samples and anchors in the $p$-th view. While a complete anchor graph can be synthesized from all views through late-fusion, this process is compromised by the inherent misalignment of anchor graphs caused by view discrepancy. This discrepancy, known as the anchor-shift problem, arises because anchors learned from $n_p$ samples differ from those learned from complete data. Consequently, varying missing samples across views lead to inconsistent anchor-shift, resulting in misaligned anchor graphs at the representation level.

To address this issue, we propose a cross-view anchor learning module. This module constructs a complete anchor graph under the assumption that each sample appears in at least one view in the incomplete multi-view scenario. Specifically, for the anchor graph $\mathbf{Z}^{(p)}$ in the $p$-th view, we update it with the following objective:

$$\min_{\mathbf{W}^{(pq)}, \mathbf{A}^{(p)}, \mathbf{Z}^{(p)}} \sum_{q=1}^{v} \left\| \mathbf{W}^{(pq)} \mathbf{X}_o^{(q)} - \mathbf{W}^{(qp)} \mathbf{A}^{(p)} \mathbf{Z}^{(p)} \mathbf{G}^{(q)} \right\|_{\mathbf{F}}^2,$$

$$\text{s.t. } \mathbf{W}^{(pq)\top} \mathbf{W}^{(pq)} = \mathbf{I}, \mathbf{Z}^{(p)\top} \mathbf{1} = \mathbf{1}, \mathbf{Z}^{(p)} \geq 0, \tag{2}$$

where $\mathbf{W}^{(pq)}$ is the projection matrix, which projects the dimension-reduced data into a higher-dimensional space. Unlike previous approaches that reduce all data to the same lower dimension, we aim to preserve high-dimensional features across views as much as possible to ensure the effectiveness of cross-view learning. A symmetric cross-view projection mechanism is designed to ensure dimensional consistency between different view pairs: if $d_p > d_q$, $\mathbf{W}^{(pq)}$ is the projection matrix that needs to be optimized; otherwise, $\mathbf{W}^{(pq)}$ equals the identity matrix, as shown below:

$$\mathbf{W}^{(pq)} = \begin{cases} \mathbf{W}^{(pq)} \in \mathbb{R}^{d_p \times d_q}, & d_p > d_q, \\ \mathbf{I} \in \mathbb{R}^{d_q \times d_q}, & d_p \leq d_q. \end{cases} \tag{3}$$

By sequentially learning the similarity between the $n_q$ points present in the $q$-th view (where $q = 1, \ldots, v$) and the $m$ anchors in the $p$-th view, we can obtain an anchor graph with a size of $m \times n$. Furthermore, the complete anchor graph can guide the learning of anchors, thereby implicitly alleviating the anchor-shift problem.

### 3.2 Affine Combination-based Reconstruction Module

The cross-view anchor learning module fills the missing columns of the anchor graph by measuring the distance between the existing samples in other views and the anchors of the current view in the same space with a projection matrix, which avoids recovering the missing samples. However, the measurement of similarity between cross-view representations is overly dependent on the reliability of the projection matrix and the consistency between views. Additionally, simply relying on cross-view information to implicitly correct anchor shifts is insufficient. Therefore, we propose to recover the incomplete data in each view to directly correct the anchors affected by the missing parts. Traditional imputation methods, which reconstruct missing data based on the similarity among all samples, require an additional quadratic space complexity, making them impractical for large-scale problems. Thus, we propose a fast reconstruction module, as follows:

$$\min_{\mathbf{X}_m^{(p)}, \mathbf{A}^{(p)}, \mathbf{Z}^{(p)}} \sum_{p=1}^{v} \left\| \left[ \mathbf{X}_o^{(p)}, \mathbf{X}_m^{(p)} \right] - \mathbf{A}^{(p)} \mathbf{Z}^{(p)} \right\|_{\mathbf{F}}^2,$$

$$\text{s.t. } \mathbf{Z}^{(p)\top} \mathbf{1} = \mathbf{1}, \mathbf{Z}^{(p)} \geq 0, \tag{4}$$

where $\mathbf{X}_o^{(p)} \in \mathbb{R}^{d_p \times n_p}$ represents the existing samples in the view, and $\mathbf{X}_m^{(p)} \in \mathbb{R}^{d_p \times n - n_p}$ represents the missing samples to be reconstructed. $\mathbf{X}_m^{(p)}$ is constructed from the anchors and their corresponding anchor graphs. The reconstructed $\mathbf{X}_m^{(p)}$ then participates in the next iteration of anchor learning, with the reconstruction of missing samples and anchor learning iterating and mutually reinforcing each other. Accurately reconstructed $\mathbf{X}_m^{(p)}$ enables the learned anchors in the next iteration to be closer to the true global anchors, whereas inaccurate reconstruction exacerbates the anchor-shift problem. However, the reconstructed missing samples are constrained within the convex hull of the anchor set in Eq. (4). According to Theorem 1, there always exist samples that cannot be reconstructed by the convex combination of anchors.

**Theorem 1.** *Suppose $\mathbf{A} = \{a_1, \ldots, a_m\}$ is an anchor set composed of cluster centers from the dataset $\mathbf{X} = \{x_1, \ldots, x_n\}$. Then, there always exist a sample $c$ belonging to $\mathbf{X}$ that lies outside the convex hull of the anchor set $\mathbf{A}$. Mathematically,*

$$\min_{\mathbf{f}} \left\| \sum_{i=1}^{m} f_i a_i - c \right\|_2^2 > 0, \exists c \in \mathbf{X},$$

$$\text{s.t. } \mathbf{f}^\top \mathbf{1} = 1, \mathbf{f} \geq 0. \tag{5}$$

When most missing samples are outside the convex hull of the anchors, the next iteration of anchor learning erroneously shifts inward, worsening anchor-shift. Therefore, we propose an affine-

combination based reconstruction strategy, which relaxes convex constraints on anchor graph rows to affine ones.

$$\min_{\mathbf{X}_m^{(p)}, \mathbf{A}^{(p)}, \mathbf{Z}^{(p)}} \sum_{p=1}^{v} \left\| \left[ \mathbf{X}_o^{(p)}, \mathbf{X}_m^{(p)} \right] - \mathbf{A}^{(p)} \mathbf{Z}^{(p)} \right\|_{\mathbf{F}}^2,$$
$$\text{s.t. } \mathbf{Z}^{(p)\top} \mathbf{1} = \mathbf{1}. \tag{6}$$

**Theorem 2.** *For a sample $c$ lying outside the convex hull of the anchor set $\mathbf{A}$, the representation constructed by the affine combination of set $\mathbf{A}$ is always closer to $c$ than that constructed by the convex combination of set $\mathbf{A}$. Mathematically,*

$$\min_{\mathbf{g}} \|\textstyle\sum_{i=1}^{m} g_i a_i - c\|_2^2 < \min_{\mathbf{f}} \|\textstyle\sum_{i=1}^{m} f_i a_i - c\|_2^2,$$
$$s.t. \ \mathbf{g}^\top \mathbf{1} = 1, \mathbf{f}^\top \mathbf{1} = 1, \mathbf{f} \geq 0. \tag{7}$$

According to Theorem 2, utilizing the affine combination of anchors facilitates the recovery of more accurate missing samples. Ultimately, by combining the cross-view anchor learning module with the affine-combination based reconstruction module, the objective of AIMC-CVR is as follows:

$$\min_{\mathbf{\Phi}} \sum_{p=1}^{v} \left\| \left[ \mathbf{X}_o^{(p)}, \mathbf{X}_m^{(p)} \right] - \mathbf{A}^{(p)} \mathbf{Z}^{(p)} \right\|_{\mathbf{F}}^2 + \beta \sum_{p=1}^{v} \left\| \mathbf{Z}^{(p)} \right\|_{\mathbf{F}}^2$$
$$+ \lambda \sum_{p=1}^{v} \sum_{q=1}^{v} \left\| \mathbf{W}^{(pq)} \mathbf{X}_o^{(q)} - \mathbf{W}^{(qp)} \mathbf{A}^{(p)} \mathbf{Z}^{(p)} \mathbf{G}^{(q)} \right\|_{\mathbf{F}}^2,$$
$$\text{s.t. } \mathbf{W}^{(pq)\top} \mathbf{W}^{(pq)} = \mathbf{I}, \mathbf{Z}^{(p)\top} \mathbf{1} = \mathbf{1}, \tag{8}$$

where $\mathbf{\Phi} = \left\{ \mathbf{X}_m^{(p)}, \mathbf{W}^{(pq)}, \mathbf{A}^{(p)}, \mathbf{Z}^{(p)} \right\}$. The hyperparameter $\beta$ helps to adjust the sparsity of the anchor graph, and $\lambda$ is a hyperparameter balancing the influence of the two modules. Finally, we concatenate the anchor graph from each view to obtain the common one, $\mathbf{Z} = \left[ \mathbf{Z}^{(1)}; \ldots; \mathbf{Z}^{(v)} \right]$, which avoids the anchor alignment problem present in other fusion methods [33]. Note that the columns of $\mathbf{Z}$ still consist of ones, ensuring that its recovered transition probability matrix $\mathbf{S}$ is a doubly stochastic matrix, as proven in the appendix. Therefore, directly performing $k$-means clustering on the left singular vectors of $\mathbf{Z}$ yields the final clustering results [26].

### 3.3  Optimization

To solve the optimization problem in Eq. (8), we propose a four step alternating iterative algorithm. When optimizing a variable, the other variables are fixed to their previous iteration values. Additionally, since each variable is independent across views, we update them sequentially by view.

**Step 1:** Update $\mathbf{X}_m^{(p)}$. Fixing $\mathbf{W}^{(pq)}, \mathbf{A}^{(p)}, \mathbf{Z}^{(p)}$ and removing terms unrelated to $\mathbf{X}_m^{(p)}$, we have the following optimization problem:

$$\min_{\mathbf{X}_m^{(p)}} \text{Tr} \left( \mathbf{X}_m^{(p)\top} \mathbf{X}_m^{(p)} - 2 \mathbf{X}_m^{(p)\top} \mathbf{A}^{(p)} \mathbf{Z}^{(p)} \mathbf{E}^{(p)} \right), \tag{9}$$

where $\mathbf{E}^{(p)} \in \{0, 1\}^{n \times n_p}$ is the index matrix, $\mathbf{E}_{ij}^{(p)} = 1$ indicates that the $i$-th sample is ranked $j$-th among the missing samples. By taking the derivative of Eq. (9) with respect to $\mathbf{X}_m^{(p)}$ and setting it to zero, we obtain the closed-form solution for $\mathbf{X}_m^{(p)}$ as follows:

$$\mathbf{X}_m^{(p)} = \mathbf{A}^{(p)} \mathbf{Z}^{(p)} \mathbf{E}^{(p)}. \tag{10}$$

The solution for $\mathbf{X}_m^{(p)}$ shows that it incorporates the anchors and anchor graphs from the previous iteration, contributing to the update of anchors in the next iteration.

**Step 2:** Update $\mathbf{W}^{(pq)}$. Fixing $\mathbf{X}_m^{(p)}$, $\mathbf{A}^{(p)}$, $\mathbf{Z}^{(p)}$ and removing terms unrelated to $\mathbf{W}^{(pq)}$, we have the following optimization problem when $d_p < d_q$:

$$\min_{\mathbf{W}^{(pq)}} \left\| \mathbf{X}_o^{(p)} - \mathbf{W}^{(pq)} \mathbf{A}^{(q)} \mathbf{Z}^{(q)} \mathbf{G}^{(p)} \right\|_{\mathbf{F}}^2 + \left\| \mathbf{W}^{(pq)} \mathbf{X}_o^{(q)} - \mathbf{A}^{(p)} \mathbf{Z}^{(p)} \mathbf{G}^{(q)} \right\|_{\mathbf{F}}^2,$$
$$\text{s.t. } \mathbf{W}^{(pq)\top} \mathbf{W}^{(pq)} = \mathbf{I}. \tag{11}$$

By further simplification, Eq. (11) can transform into the following form:

$$\max_{\mathbf{W}^{(pq)}} \text{Tr}\left( \mathbf{W}^{(pq)\top} \mathbf{B}^{(pq)} \right),$$
$$\text{s.t. } \mathbf{W}^{(pq)\top} \mathbf{W}^{(pq)} = \mathbf{I}. \tag{12}$$

where $\mathbf{B}^{(pq)} = \mathbf{X}_o^{(p)} (\mathbf{A}^{(q)} \mathbf{Z}^{(q)} \mathbf{G}^{(p)})^\top + \mathbf{A}^{(p)} \mathbf{Z}^{(p)} \mathbf{G}^{(q)} \mathbf{X}_o^{(p)\top}$. According to reference [43], the optimal $\mathbf{W}^{(pq)}$ is derived from the product of the left and right singular vectors of $\mathbf{B}^{(pq)}$.

**Step 3:** Update $\mathbf{A}^{(p)}$. Fixing $\mathbf{X}_m^{(p)}$, $\mathbf{W}^{(pq)}$, $\mathbf{Z}^{(p)}$ and removing terms unrelated to $\mathbf{A}^{(p)}$, we have the following optimization problem:

$$\min_{\mathbf{A}^{(p)}} \text{Tr}\left( \mathbf{A}^{(p)} \mathbf{C}^{(p)} \mathbf{A}^{(p)\top} - 2\mathbf{A}^{(p)\top} \mathbf{D}^{(p)} \right), \tag{13}$$

where $\mathbf{C}^{(p)} = \mathbf{Z}^{(p)} \mathbf{Z}^{(p)\top} + \lambda \sum_{q=1}^v \mathbf{Z}^{(p)} \mathbf{G}^{(q)} \mathbf{G}^{(q)\top} \mathbf{Z}^{(p)\top}$, $\mathbf{D}^{(p)} = \mathbf{X}^{(p)} \mathbf{Z}^{(p)\top} + \lambda \sum_{q=1}^v \mathbf{W}^{(qp)\top} \mathbf{W}^{(pq)} \mathbf{X}_o^{(q)} \mathbf{G}^{(q)\top} \mathbf{Z}^{(p)\top}$. By taking the derivative of Eq. (13) with respect to $\mathbf{A}^{(p)}$ and setting it to zero, we obtain the closed-form solution for $\mathbf{A}^{(p)}$ as follows:

$$\mathbf{A}^{(p)} = \mathbf{D}^{(p)} \mathbf{C}^{(p)-1}. \tag{14}$$

**Step4:** Update $\mathbf{Z}^{(p)}$. Since each column of $\mathbf{Z}^{(p)}$ is independent of others, we optimize the $i$-th column $\mathbf{z}_i^{(p)}$ of $\mathbf{Z}^{(p)}$ while fixing $\mathbf{X}_m^{(p)}$, $\mathbf{W}^{(pq)}$, $\mathbf{A}^{(p)}$ as follows:

$$\min_{\mathbf{z}_i^{(p)}} \mathbf{z}_i^{(p)\top} \mathbf{H} \mathbf{z}_i^{(p)} - 2\mathbf{t}_i^\top \mathbf{z}_i^{(p)},$$
$$\text{s.t. } \mathbf{z}_i^{(p)\top} \mathbf{1} = 1. \tag{15}$$

where $\mathbf{H} = (\lambda \sum_{q=1}^v \sigma_i^{(q)} + 1) \mathbf{A}^{(p)\top} \mathbf{A}^{(p)} + \beta \mathbf{I}$, $\mathbf{t}_i = \lambda \sum_{q=1}^v \sigma_i^{(q)} \mathbf{X}_{:,i}^{(q)\top} \mathbf{W}^{(pq)\top} \mathbf{W}^{(qp)} \mathbf{A}^{(p)} + \mathbf{X}_{:,i}^{(p)\top} \mathbf{A}^{(p)}$. When $i$-th sample exists in the $p$-th view $\sigma_i^{(q)} = 1$, else $\sigma_i^{(q)} = 0$. We employ the Lagrange multiplier method to tackle the above problem. Firstly, the Lagrangian function for Eq. (15) is as follows:

$$\mathbf{L} = \mathbf{z}_i^{(p)\top} \mathbf{H} \mathbf{z}_i^{(p)} - 2\mathbf{t}_i^\top \mathbf{z}_i^{(p)} + \alpha_i (\mathbf{z}_i^{(p)\top} \mathbf{1} - 1), \tag{16}$$

where $\alpha_i$ is the Lagrangian multiplier. The corresponding KTT conditions is

$$\begin{cases} \mathbf{H} \mathbf{z}_i^{(p)} - \mathbf{t}_i + \frac{\alpha_i}{2} \mathbf{1} = 0, \\ \mathbf{z}_i^{(p)\top} \mathbf{1} = 1. \end{cases} \tag{17}$$

By substituting the first term into the second, we can get $\alpha_i = 2 \frac{(\mathbf{H}^{-1} \mathbf{t}_i)^\top \mathbf{1} - 1}{\mathbf{1}^\top \mathbf{H}^{-1} \mathbf{1}}$. Then we have $\mathbf{z}_i^{(p)} = \mathbf{H}^{-1}(\mathbf{t}_i - \frac{\alpha}{2} \mathbf{1})$. The entire optimization procedure for AIMC-CVR is summarized in Algorithm 1.

---

**Algorithm 1** The proposed AIMC-CVR

---

**Input:** Multi-view dataset $\left\{\mathbf{X}^{(p)}\right\}_{p=1}^{v}$, anchors number $m$, clusters number $k$, parameters $\beta$, $\lambda$.

1: Initialize $\left\{\mathbf{A}^{(p)}\right\}_{p=1}^{v}$, $\left\{\mathbf{W}^{(pq)}\right\}_{p,q=1}^{v}$ and $\left\{\mathbf{Z}^{(p)}\right\}_{p=1}^{v}$
2: **while** not converged **do**
3:     **for** $p = 1 \rightarrow v$ **do**
4:         Update $\mathbf{X}_m^{(p)}$ with Eq. (10).
5:         **for** $q = 1 \rightarrow v$ **do**
6:            Update $\mathbf{W}^{(pq)}$ by solving Eq. (12).
7:         **end for**
8:         Update $\mathbf{A}^{(p)}$ with Eq. (14).
9:         Update $\mathbf{Z}^{(p)}$ by solving Eq. (15).
10:     **end for**
11: **end while**
12: Concatenate $\mathbf{Z}^{(p)}$ to obtain $\mathbf{Z}$.
**Output:** Performing $k$-means on $\mathbf{Z}$ to get the final cluster results.

---

### 3.4 Complexity Analysis

The computational complexity of AIMC-CVR primarily consists of solving four variables. When updating $\mathbf{X}_m^{(p)}$, the complexity of matrix multiplication is $\mathcal{O}(nmd_p)$. For updating $\mathbf{W}^{(pq)}$, the complexity of matrix multiplication is $\mathcal{O}(nmd_p + nd_p{}^2 + d_p{}^3)$, and the SVD decomposition complexity is $\mathcal{O}(d_p{}^3)$. When optimizing $\mathbf{A}^{(p)}$, the complexity of matrix multiplication is $\mathcal{O}(nmd_p + m^2 d_p + nd_p d_q)$, and the complexity of inversion is $\mathcal{O}(m^3)$. When optimizing $\mathbf{Z}^{(p)}$, the complexity of matrix multiplication is $\mathcal{O}(m^2 d_p + nmd_p + nd_p d_q)$. Therefore, in each iteration, AIMC-CVR consumes a time complexity of $\mathcal{O}(n \sum_{p=1}^{v}(md_p + d_p{}^2 + d_p \sum_{q=1}^{v} d_q) + \sum_{p=1}^{v}(m^2 d_p + d_p^3))$, which is linear with respect to $n$.

The space complexity of AIMC-CVR is $\mathcal{O}(n(m + \sum_{p=1}^{v} d_p) + m \sum_{p=1}^{v} d_p + \sum_{p=1}^{v} \sum_{q=1}^{v} d_p d_q)$, primarily stemming from storing relevant matrix variables, also scales linearly with the number of samples.

### 3.5 Convergence Analysis

In this section, we provide a theoretical analysis of the convergence of our proposed AIMC-CVR. The objective function in Eq. (8) is non-convex when considering all variables simultaneously. To address this, we employ a four-step iterative optimization algorithm, detailed in Algorithm 1, where each variable is optimized sequentially while keeping the others fixed. During each iteration, the variables being optimized have analytical solutions, ensuring that the objective function of AIMC-CVR decreases monotonically with successive iterations. Furthermore, since the objective function in Eq. (8) is bounded below by zero, our proposed AIMC-CVR is guaranteed to converge to a local minimum.

## 4 Experiments

### 4.1 Experimental Setup

**Datasets description.** Seven widely used multi-view datasets are employed to evaluate the performance of AIMC-CVR, including: MSRCV [44] is composed of images from seven categories. WebKB[2] contains text and citations collected from the website. Wiki [45] is a dual-view dataset with text-image pairs. Hdigit[3] is a dataset composed of handwritten digit images. YTF10 and YTF20

---

[2]http://www.cs.umd.edu/sen/lbc-proj/LBC.html
[3]https://cs.nyu.edu/%7Eroweis/data.html

are two subsets extracted from the YouTubeFace[4] dataset. MNIST[5] is a subset extracted from a larger dataset supplied by NIST. Details of these datasets are list in the Table 1. Following [26], we randomly remove 10% to 90% of the samples in 10% intervals to create missing versions of the above datasets, which ensures each sample exists in at least one view.

Table 1: Employed datasets in experiments.

| Datasets | #Samples (n) | #Views (v) | #Clusters (k) | #Dimensionality (d_p) | | | |
|----------|--------------|------------|---------------|------|------|------|-----|
| MSRCV | 210 | 3 | 7 | 256 | 512 | 1302 | - |
| WebKB | 1051 | 2 | 2 | 334 | 2949 | - | - |
| Wiki | 2866 | 2 | 10 | 10 | 128 | - | - |
| Hdigit | 10000 | 2 | 10 | 256 | 784 | - | - |
| YTF10 | 38654 | 4 | 10 | 512 | 576 | 640 | 944 |
| YTF20 | 63896 | 4 | 20 | 512 | 576 | 640 | 944 |
| MNIST | 70000 | 4 | 9 | 512 | 576 | 640 | 944 |

**Compared methods.** We compared our proposed method with the following eight state-of-the-art incomplete multi-view clustering algorithms: DAIMC[22], UEAF[23], EEIMVC[19], FLSD[24], V³H[15], IMVC-CBG[26], SCBGL[29], DVSAI[31]. The first five comparison methods are traditional similarity-based approaches, while the latter three are large-scale anchor-based methods.

**Implementation details.** For all comparison algorithms, we set the parameters according to their descriptions in the corresponding literature. In our method, the anchors number is searched in $[k, 2k, 3k]$, and the parameter $\lambda$ and $\beta$ are both searched in $[0.001, 0.01, 0.1, 1, 10, 100]$. To evaluate the clustering performance, we employ accuracy (ACC), normalized mutual information (NMI), Purity, and Fscore for comparison. Each compared algorithm was tested across datasets with different missing rates, and the results for each missing rate were averaged to obtain the clustering results. Additionally, we conduct 20 repetitions of the $k$-means step for all algorithms and calculate the mean and variance for the final experimental result. All experiments were conducted on a desktop computer equipped with an Intel Core i9-10900X CPU, 64GB of RAM.

## 4.2 Clustering Performance Comparison

We compare AIMC-CVR with eight state-of-the-art algorithms across seven datasets, as shown in Table 2. Our algorithm demonstrates superior or competitive clustering performance across all datasets, highlighting its effectiveness. On smaller datasets such as MSRCV, WebKB, Wiki, and Hdigit, our method achieves the highest ACC in all cases, surpassing the second-best algorithms by 3.65%, 7.89%, 0.19% and 3.96%. This showcases our algorithm's robustness and efficacy in handling incomplete multi-view datasets. Furthermore, Fig. 2 shows the clustering accuracy (ACC) curves for all algorithms across different datasets as the missing rate varies. Our algorithm consistently outperforms all others at every missing rate, demonstrating its superiority.

Additionally, our method demonstrates excellent scalability, successfully processing three large-scale datasets that the traditional methods cannot handle due to memory constraints. By addressing the anchor-shift problem, which was overlooked by other three anchor-based methods (IMVC-CBG, SCBGL, DVSAI), we achieved a notable enhancement in clustering performance. Specifically, on YTF10, YTF20, and MNIST datasets, our algorithm achieves ACC improvements over the second-best algorithms by 3.18%, 0.42% and 7.47%.

## 4.3 Ablation Study

To showcase the effectiveness of different modules and strategies, we constructed five variants of AIMC-CVR as follows: (1) AIMC-CVR-v1 removes the cross-view anchor learning module by setting $\lambda = 0$. (2) AIMC-CVR-v2 removes the affine combination-based reconstruction module. (3) AIMC-CVR-v3 removes the sparsity regularization term by setting $\beta = 0$. (4) AIMC-CVR-v4 keeps the initialized $\mathbf{A}^{(p)}$ fixed and does not update it during subsequent optimization. (5) AIMC-CVR-v5 replaces affine combination with convex combination by adding non-negative constraints to $\mathbf{Z}^{(p)}$.

---

[4]https://www.micc.unifi.it/resources/datasets/e-ytf/
[5]http://yann.lecun.com/exdb/mnist/

Table 2: Clustering performance on seven datasets. '-' means unable to run due to memory limitation.

| Datasets | DAIMC | UEAF | EEIMVC | FLSD | V³H | IMVC-CBG | SCBGL | DVSAI | Proposed |
|---|---|---|---|---|---|---|---|---|---|
| | | | | | ACC(%) | | | | |
| MSRCV | 66.04±7.40 | 55.47±4.67 | 69.16±4.87 | 58.54±6.38 | 71.11±5.42 | 63.51±1.73 | 69.94±0.70 | 79.87±0.10 | **83.52±1.00** |
| WebKB | 78.36±5.77 | 82.64±0.67 | 61.67±3.49 | 78.21±0.00 | 74.69±10.53 | 83.48±0.14 | 75.21±0.02 | 68.64±0.00 | **91.37±0.00** |
| Wiki | 46.07±1.11 | 47.75±0.05 | 48.45±0.14 | 48.40±0.30 | 32.57±0.46 | 47.27±0.02 | 46.03±0.07 | 35.81±0.13 | **48.64±0.15** |
| Hdigit | 56.13±4.46 | 65.01±3.34 | 58.35±3.00 | 63.91±4.32 | 78.33±6.38 | 62.36±0.08 | 68.03±0.10 | 76.11±0.00 | **82.29±0.01** |
| YTF10 | - | - | - | - | - | 70.46±0.32 | 80.55±1.77 | 78.38±0.00 | **83.73±1.29** |
| YTF20 | - | - | - | - | - | 70.74±0.99 | 76.56±2.38 | 76.00±0.33 | **76.98±2.35** |
| MNIST | - | - | - | - | - | 53.24±0.30 | 60.58±0.00 | 61.95±0.00 | **69.42±0.01** |
| | | | | | NMI(%) | | | | |
| MSRCV | 58.58±5.05 | 46.31±2.81 | 57.38±3.17 | 52.89±3.24 | 64.84±3.25 | 56.25±2.12 | 59.90±0.98 | 69.31±0.16 | **72.17±1.20** |
| WebKB | 18.29±10.60 | 23.52±0.97 | 3.57±0.56 | 4.56±4.14 | 21.68±11.13 | 21.92±0.01 | 16.60±0.00 | 16.44±0.00 | **51.65±0.00** |
| Wiki | 32.31±0.83 | **43.68±0.03** | 39.71±0.11 | 40.01±0.15 | 19.03±0.21 | 35.91±0.02 | 33.55±0.07 | 19.35±0.10 | 39.78±0.17 |
| Hdigit | 47.26±2.82 | 55.04±1.48 | 47.58±1.32 | 57.94±1.60 | **81.05±2.18** | 51.33±0.11 | 54.17±0.05 | 62.94±0.00 | 67.05±0.02 |
| YTF10 | - | - | - | - | - | 73.22±0.81 | 81.58±0.80 | **83.79±0.01** | 81.81±0.79 |
| YTF20 | - | - | - | - | - | 76.25±0.76 | 80.24±0.91 | 83.15±0.12 | **83.20±0.97** |
| MNIST | - | - | - | - | - | 50.59±0.01 | 57.62±0.01 | 55.86±0.02 | **61.68±0.00** |
| | | | | | Purity(%) | | | | |
| MSRCV | 68.44±6.10 | 57.37±3.99 | 70.62±3.88 | 61.19±5.17 | 73.66±4.05 | 65.32±1.47 | 71.24±0.98 | 79.87±0.1 | **83.52±1.00** |
| WebKB | 81.75±3.68 | 82.91±0.09 | 78.12±0.00 | 78.72±1.15 | 81.38±3.25 | 83.48±0.14 | 78.12±0.00 | 79.03±0.00 | **91.37±0.00** |
| Wiki | 49.62±0.85 | 51.71±0.04 | 50.26±0.12 | 51.71±0.11 | 38.43±0.37 | 51.42±0.02 | 50.39±0.08 | 38.77±0.11 | **51.88±0.08** |
| Hdigit | 58.89±3.79 | 66.97±2.50 | 60.60±1.93 | 66.90±2.77 | 81.37±4.72 | 64.17±0.09 | 68.34±0.10 | 76.39±0.00 | **82.29±0.01** |
| YTF10 | - | - | - | - | - | 75.29±0.16 | 83.72±0.91 | 82.83±0.00 | **84.43±1.21** |
| YTF20 | - | - | - | - | - | 75.40±1.04 | 80.12±1.38 | 81.23±0.21 | **81.27±1.81** |
| MNIST | - | - | - | - | - | 58.06±0.01 | 64.51±0.02 | 62.95±0.00 | **71.88±0.00** |
| | | | | | Fscore(%) | | | | |
| MSRCV | 54.64±5.72 | 42.63±2.82 | 55.46±3.80 | 47.45±3.59 | 60.45±4.08 | 50.05±2.19 | 56.36±1.02 | 66.18±0.17 | **70.77±1.22** |
| WebKB | 76.31±5.86 | 78.51±0.73 | 62.75±2.14 | 79.40±0.00 | 71.82±0.00 | 82.28±0.11 | 69.02±0.01 | 64.38±0.01 | **87.76±0.00** |
| Wiki | 35.05±0.98 | 36.00±0.10 | **38.65±0.13** | 37.57±0.18 | 22.29±0.20 | 36.49±0.02 | 34.98±0.09 | 23.59±0.10 | 37.05±0.08 |
| Hdigit | 44.48±3.29 | 52.52±2.50 | 45.21±1.68 | 53.6±2.61 | **75.85±4.62** | 48.75±0.30 | 52.96±0.05 | 63.10±0.00 | 69.19±0.02 |
| YTF10 | - | - | - | - | - | 64.40±0.04 | 77.02±1.10 | 77.27±0.00 | **77.82±1.03** |
| YTF20 | - | - | - | - | - | 57.42±1.17 | 69.55±1.86 | **72.38±0.36** | 70.97±2.35 |
| MNIST | - | - | - | - | - | 44.14±0.20 | 51.75±0.01 | 50.56±0.02 | **58.86±0.00** |

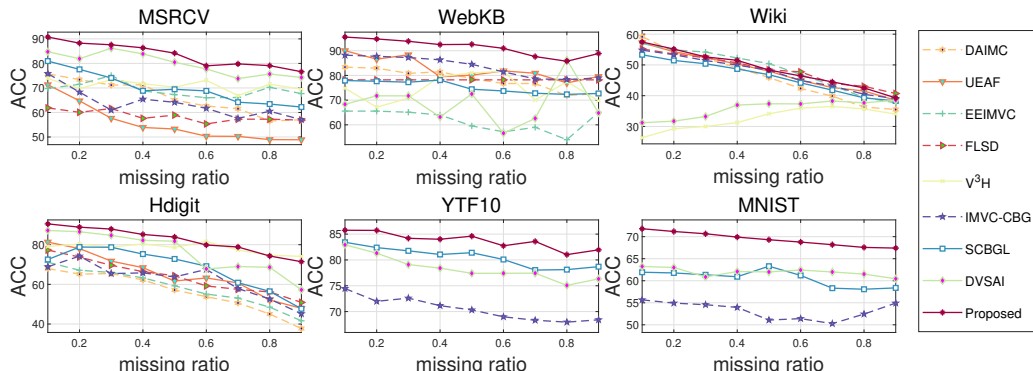

Figure 2: The curves for ACC across different datasets as the missing rate varies.

The comparison results between the above variants and our method are shown in Table 3. Both modules in AIMC-CVR are essential, the absence of either one leads to decreased clustering performance as shown in AIMC-CVR-v1 and AIMC-CVR-v2. Compared with AIMC-CVR-v3, appropriate constraints on the sparsity of $\mathbf{Z}^{(p)}$ can prevent excessively divergent values. AIMC-CVR-v4 uses initial anchors learned from single-view data with missing samples, leading to anchor-shift problem, while our method effectively mitigates it and enhances clustering results. Compared to AIMC-CVR-v5, our method achieves better performance by reconstructing missing samples with affine combination of anchors, which leads to more accurate learning of anchors and anchor graphs.

## 4.4 Convergence Study

Fig. 3 illustrates the objective value of our proposed algorithm against the number of iterations on four datasets. The algorithm shows rapid convergence within the first 50 iterations for all datasets. The objective value decreases significantly in the initial iterations and gradually stabilizes, indicating efficient attainment of an optimal or near-optimal solution. This rapid convergence is especially notable on the MSRCV and WebKB datasets, where the objective value plateaus before 50 iterations.

The consistent convergence patterns across different datasets validate the robustness and efficiency of our algorithm in handling incomplete multi-view clustering tasks.

Table 3: Ablation studies of AIMC-CVR with different variants.

| Methods | MSRCV | WebKB | Wiki | Hdigit | YTF10 | YTF20 | MNIST |
|---------|-------|-------|------|--------|-------|-------|-------|
| | | | ACC(%) | | | | |
| AIMC-CVR-v1 | 67.80±0.92 | 72.77±0.00 | 45.38±0.10 | 54.01±0.09 | 80.31±1.21 | 62.64±1.75 | 59.80±0.01 |
| AIMC-CVR-v2 | 71.95±0.75 | 72.18±0.00 | 46.16±0.09 | 67.61±0.02 | 78.86±0.53 | 75.01±2.06 | 68.35±0.01 |
| AIMC-CVR-v3 | 44.41±1.89 | 87.47±0.00 | 28.67±0.28 | 29.53±0.47 | 63.47±0.75 | 65.65±2.08 | 57.10±0.01 |
| AIMC-CVR-v4 | 66.23±1.50 | 82.82±0.00 | 46.23±0.15 | 27.66±0.46 | 69.97±0.40 | 71.84±1.99 | 55.95±0.35 |
| AIMC-CVR-v5 | 71.92±0.73 | 72.15±0.00 | 46.16±0.10 | 67.63±0.03 | 78.96±0.57 | 75.12±2.08 | 68.35±0.01 |
| AIMC-CVR | **83.52±1.00** | **91.37±0.00** | **48.64±0.15** | **82.29±0.01** | **83.73±1.29** | **76.98±2.35** | **69.42±0.01** |

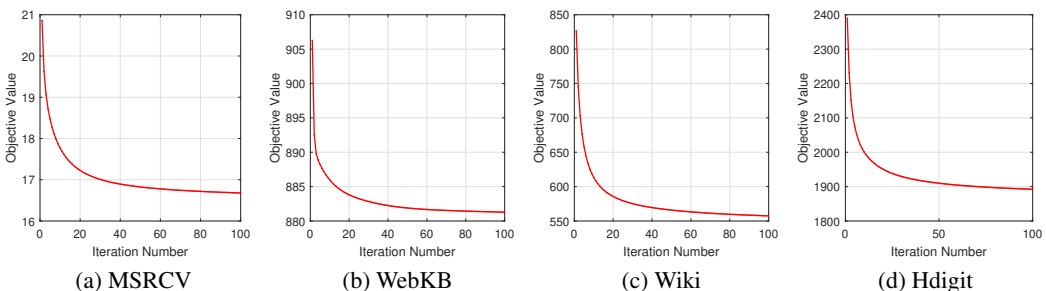

(a) MSRCV      (b) WebKB      (c) Wiki      (d) Hdigit

Figure 3: Variation of the objective value with increasing iteration number on four datasets.

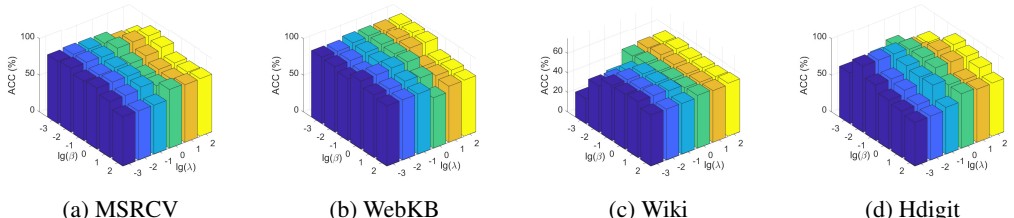

(a) MSRCV      (b) WebKB      (c) Wiki      (d) Hdigit

Figure 4: Sensitivity analysis of $\beta$ and $\lambda$ on four datasets.

## 4.5 Parameter Analysis

AIMC-CVR has two main hyperparameters: $\beta$ controls the sparsity of the anchor graph and $\lambda$ controls the weight of the cross-view learning module. We test the performance of AIMC-CVR with different combinations of these hyperparameters, as shown in Fig. 4. The impact of these parameters varies across datasets, but values of $\beta$ and $\lambda$ close to 1 generally perform well. Experimental results shows that the cross-view learning module is crucial for constructing complete anchor graphs, while $\beta$ ensures the graph to be not too sparse. Proper parameter settings lead to good clustering results.

## 5 Conclusion

In this paper, we introduce an AIMC-CVR method to alleviate the anchor-shift problem in anchor-based incomplete multi-view clustering. Additionally, we explore the blind spots in sample reconstruction with affine combination. Experiments and theoretical analysis validate the effectiveness of the proposed AIMC-CVR method.

## Acknowledgment

This work is supported by National Science and Technology Innovation 2030 Major Project under Grant No. 2022ZD0209103, the National Science Fund for Distinguished Young Scholars of China (No. 62325604), and the National Natural Science Foundation of China (No. 62276271, 62476281 and 62406329).

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

# A    Appendix

## A.1    Limitation

The primary limitation of AIMC-CVR lies in its difficulty in handling high-dimensional data. Although the designed symmetric cross-view projection mechanism help to preserve high-dimensional features, it also introduces additional time complexity of $\mathcal{O}(d_p{}^3)$ and space complexity of $\mathcal{O}(d_p d_q)$. A direction for future research is the design of a well-structured unified cross-view metric space to address high-dimensional data. Moreover, different levels of confidence should be assigned to relationships between different views in cross-view learning. Incorporating the interrelations among views into cross-view learning can help to improve AIMC-CVR.

## A.2    Proof of Theorem 1

To prove Theorem 1, we first present two lemmas along with their corresponding proofs.

**Lemma 1.** *If the set of anchors $\mathcal{A} = \{a_1, \ldots, a_m\}$ contains distinct points and for any anchor $a_i$, $a_i$ is not a vertex of the convex hull of $\mathcal{A}$, i.e., $a_i \in conv(\mathcal{A} \setminus \{a_i\})$ where $conv(\cdot)$ denotes the convex hull, then $\mathcal{A}$ must be an empty set.*

*Proof.* When $j = 1$, we have $a_1 \in conv(\mathcal{A} \setminus \{a_1\})$, then

$$
\begin{aligned}
conv(\mathcal{A}\setminus\{a_1\}) &= conv(conv(\mathcal{A}\setminus\{a_1\})) \\
&= conv(conv(\mathcal{A}\setminus\{a_1\}) \cup \{a_1\}) \\
&\supseteq conv((\mathcal{A}\setminus\{a_1\}) \cup \{a_1\}) \\
&= conv(\mathcal{A}).
\end{aligned}
\tag{18}
$$

Since $\mathcal{A}\setminus\{a_1\} \subseteq \mathcal{A}$, then

$$
conv(\mathcal{A}\setminus\{a_1\}) \subseteq conv(\mathcal{A}).
\tag{19}
$$

Along with Eq. (18), we have

$$
conv(\mathcal{A}\setminus\{a_1\}) = conv(\mathcal{A}).
\tag{20}
$$

Assuming that for $1 \leq j < m$, we have

$$
conv(\mathcal{A}\setminus \cup_{i=1}^{j} \{a_i\}) = conv(\mathcal{A}).
\tag{21}
$$

Based on $a_{j+1} \in conv(\mathcal{A}\setminus\{a_{j+1}\})$, $a_{j+1}$ is not a vertex of $conv(\mathcal{A})$. Therefore, $a_{j+1}$ is not a vertex of $conv(\mathcal{A}\setminus \cup_{i=1}^{j} \{a_i\})$. Then we have

$$
\begin{aligned}
a_{j+1} &\in conv(\mathcal{A}\setminus \cup_{i=1}^{j} \{a_i\}\setminus\{a_{j+1}\}) \\
&= conv(\mathcal{A}\setminus \cup_{i=1}^{j+1} \{a_i\}).
\end{aligned}
\tag{22}
$$

Therefore,

$$
\begin{aligned}
conv(\mathcal{A}\setminus \cup_{i=1}^{j+1} \{a_i\}) &= conv(conv(\mathcal{A}\setminus \cup_{i=1}^{j+1} \{a_i\})) \\
&= conv(conv(\mathcal{A}\setminus \cup_{i=1}^{j+1} \{a_i\}) \cup \{a_{j+1}\}), \\
&\supseteq conv((\mathcal{A}\setminus \cup_{i=1}^{j+1} \{a_i\}) \cup \{a_{j+1}\}) \\
&= conv((\mathcal{A}\setminus \cup_{i=1}^{j} \{a_i\})) \\
&= conv(\mathcal{A}).
\end{aligned}
\tag{23}
$$

In summary, we have

$$
\begin{aligned}
conv(\mathcal{A}\setminus \cup_{i=1}^{m} \{a_i\}) &= conv(\mathcal{A}), \\
conv(\varnothing) &= conv(\mathcal{A}), \\
\mathcal{A} &= \varnothing.
\end{aligned}
\tag{24}
$$

This completes the proof.  □

**Lemma 2.** *If the set of anchors $\mathcal{A} = \{a_1, \dots, a_m\}$ contains distinct anchors and $\mathcal{A} \neq \varnothing$, then there exists an anchor $a_i$ such that $a_i$ is a vertex of the convex hull $conv(\mathcal{A})$. In other words, $a_i \notin conv(\mathcal{A}\backslash\{a_i\})$.*

*Proof.* Assume that for any anchor $a_i$, we have $a_i \in conv(\mathcal{A}\backslash\{a_i\})$. According to Lemma 1, we have $\mathcal{A} = \varnothing$, which contradicts the given condition. Therefore, the assumption is false. There exists an anchor $a_i$ such that $a_i$ is a vertex of the convex hull $conv(\mathcal{A})$. In other words, $a_i \notin conv(\mathcal{A}\backslash\{a_i\})$. This completes the proof. $\qquad\square$

Based on Lemma 1 and Lemma 2, we provide the proof of Theorem 1 as follows:

*Proof.* Based on Lemma 2, there exists an anchor $a_i$ such that $a_i$ is a vertex of the convex hull $conv(\mathcal{A})$. Assume $\mathcal{X}_{a_i} \subseteq conv(\mathcal{A})$. Since $a_i$ is a vertex of $conv(\mathcal{A})$, we have $a_i \notin conv(conv(\mathcal{A}\backslash\{a_i\}))$. Given $\mathcal{X}_{a_i} \subseteq conv(\mathcal{A})$, we have:

$$
\begin{aligned}
\mathcal{X}_{a_i}\backslash\{a_i\} &\subseteq conv(\mathcal{A})\backslash\{a_i\}, \\
conv(\mathcal{X}_{a_i}\backslash\{a_i\}) &\subseteq conv(conv(\mathcal{A}\backslash\{a_i\})).
\end{aligned}
\tag{25}
$$

Then

$$
a_i \notin conv(\mathcal{X}_{a_i}\backslash\{a_i\}),
\tag{26}
$$

which contradicts the fact that $a_i$ is the cluster center of $\mathcal{X}_{a_i}$. Thus, the assumption does not hold, and $\mathcal{X}_{a_i} \not\subseteq conv(\mathcal{A})$. There exists $c \in \mathcal{X}_{a_i} \subseteq \mathcal{X}$ such that

$$
\min_{\mathbf{f}} \left\| \sum_{i=1}^{m} f_i a_i - c \right\|_2^2 > 0, \text{ s.t. } \mathbf{f}^\top \mathbf{1} = 1, \mathbf{f} \geq 0.
\tag{27}
$$

This completes the proof. $\qquad\square$

### A.3 Proof of Theorem 2

*Proof.* We denote $\mathbf{f}^*$ and $\mathbf{g}^*$ as the optimal values of $\mathbf{f}$ and $\mathbf{g}$ in Eq. (7). According to the Pythagorean theorem, we have

$$
\left\| \sum_{i=1}^{m} g_i^* a_i - c \right\|_2^2 + \left\| \sum_{i=1}^{m} g_i^* a_i - \sum_{i=1}^{m} f_i^* a_i \right\|_2^2 = \left\| \sum_{i=1}^{m} f_i^* a_i - c \right\|_2^2.
\tag{28}
$$

In the projection space $aff(\mathcal{A})$ where $aff(\cdot)$ denotes the affine hull, Theorem 1 still holds. Therefore, we have

$$
\left\| \sum_{i=1}^{m} g_i^* a_i - \sum_{i=1}^{m} f_i^* a_i \right\|_2^2 > 0.
\tag{29}
$$

Then

$$
\left\| \sum_{i=1}^{m} g_i^* a_i - c \right\|_2^2 < \left\| \sum_{i=1}^{m} f_i^* a_i - c \right\|_2^2.
\tag{30}
$$

This completes the proof. $\qquad\square$

### A.4 Proof of Doubly Stochastic Matrix S

**Lemma 3.** *Let $\mathbf{Z}$ be an $m \times n$ matrix where the sum of each column is 1. Define $\mathbf{Q}$ as a diagonal matrix where the $i$-th diagonal element is the sum of the elements in the $i$-th row of Z. Then the matrix $\mathbf{S} = \mathbf{Z}^\top \mathbf{Q}^{-1} \mathbf{Z}$ is doubly stochastic, meaning that each row and each column of sums to 1.*

*Proof.* Denote $\mathbf{S}_{ij}$ to be the $j$-th element in the $i$-th row of $\mathbf{S}$, $\mathbf{q}_l$ to be $i$-th diagonal element of $\mathbf{Q}$, we can derive that

$$
\begin{aligned}
\sum_{j=1}^{n} \mathbf{S}_{ij} &= \sum_{j=1}^{n} \sum_{l=1}^{m} \mathbf{Z}_{li} \frac{1}{\mathbf{q}_l} \mathbf{Z}_{lj} \\
&= \sum_{l=1}^{m} \frac{1}{\mathbf{q}_l} \mathbf{Z}_{li} \sum_{j=1}^{n} \mathbf{Z}_{lj} \\
&= \sum_{l=1}^{m} \frac{1}{\mathbf{q}_l} \mathbf{Z}_{li} \mathbf{q}_l \\
&= \sum_{l=1}^{m} \mathbf{Z}_{li} \\
&= 1.
\end{aligned}
\tag{31}
$$

Similarly, it can be proven that $\sum_{i=1}^{n} \mathbf{S}_{ij} = 1$. This completes the proof. $\qquad\square$

### A.5 Convergence Study on More Datasets

In Fig. 5, we further plot the curve of the objective function values against the number of iterations for the proposed algorithm on the YTF10, YTF20, and MNIST datasets. It can be observed that the objective values monotonically decrease with the increasing number of iterations on these three datasets, gradually approaching stability, thus confirming the convergence of AIMC-CVR.

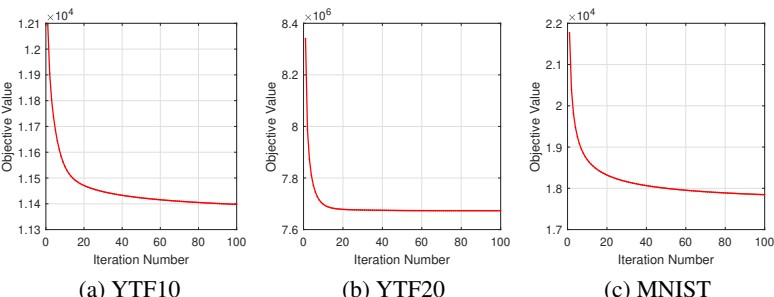

(a) YTF10      (b) YTF20      (c) MNIST

Figure 5: Variation of the objective value with increasing iteration number on other three datasets.

### A.6 Parameter Analysis on More Datasets

In Fig. 6, we further plot the variation of clustering performance of the proposed algorithm on the YTF10, YTF20, and MNIST datasets with different values of two hyperparameters, $\beta$ and $\lambda$. It can be observed that AIMC-CVR exhibits less sensitivity to parameters on the YTF10 and YTF20 datasets. However, on the MNIST dataset, higher values of $\lambda$ lead to better clustering performance.

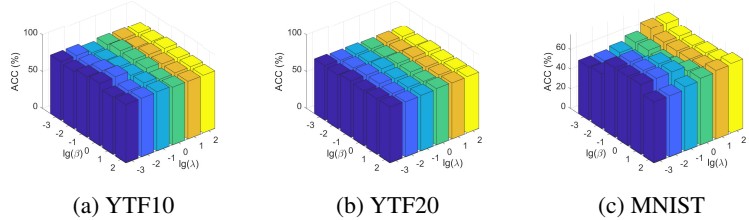

(a) YTF10      (b) YTF20      (c) MNIST

Figure 6: Sensitivity analysis of $\beta$ and $\lambda$ on other three datasets.

## A.7 Ablation Study on More Metrics

To further demonstrate the effectiveness of our approach, we compared our algorithm with five variants and three other clustering metrics in Table 4. Compared to the other variants, AIMC-CVR exhibits better NMI, Purity, and Fscore on all datasets, validating the effectiveness of our method.

Table 4: Ablation studies of AIMC-CVR with different variants on Other Metrics.

| Methods | MSRCV | WebKB | Wiki | Hdigit | YTF10 | YTF20 | MNIST |
|---------|-------|-------|------|--------|-------|-------|-------|
| NMI(%) | | | | | | | |
| AIMC-CVR-v1 | 59.26±1.14 | 20.65±0.00 | 35.47±0.13 | 47.33±0.05 | 81.06±0.84 | 68.43±0.91 | 52.40±0.01 |
| AIMC-CVR-v2 | 61.66±0.83 | 18.29±0.01 | 35.70±0.08 | 56.50±0.02 | 79.89±0.39 | 79.99±0.86 | 60.94±0.01 |
| AIMC-CVR-v3 | 30.12±1.67 | 46.16±0.02 | 11.86±0.19 | 14.18±0.21 | 64.96±0.53 | 70.64±0.78 | 57.51±0.00 |
| AIMC-CVR-v4 | 53.33±1.65 | 39.77±0.00 | 35.50±0.17 | 12.60±0.18 | 67.44±0.26 | 75.92±0.62 | 50.73±0.15 |
| AIMC-CVR-v5 | 61.57±0.81 | 18.32±0.00 | 35.59±0.08 | 56.51±0.03 | 79.96±0.33 | 80.10±0.74 | 60.94±0.01 |
| AIMC-CVR | **72.17±1.20** | **51.65±0.00** | **39.78±0.17** | **67.05±0.02** | **81.81±0.79** | **83.20±0.97** | **61.68±0.00** |
| Purity(%) | | | | | | | |
| AIMC-CVR-v1 | 70.03±0.84 | 79.46±0.00 | 47.94±0.09 | 58.56±0.07 | 83.29±0.94 | 68.32±1.33 | 61.43±0.01 |
| AIMC-CVR-v2 | 73.52±0.75 | 79.12±0.00 | 48.71±0.11 | 71.17±0.02 | 82.49±0.49 | 80.37±1.51 | 71.10±0.01 |
| AIMC-CVR-v3 | 45.78±1.8 | 89.48±0.00 | 31.31±0.28 | 31.06±0.45 | 68.78±0.66 | 70.66±1.32 | 62.07±0.01 |
| AIMC-CVR-v4 | 66.85±1.43 | 87.47±0.00 | 48.36±0.14 | 29.06±0.43 | 73.34±0.42 | 76.06±1.12 | 59.22±0.17 |
| AIMC-CVR-v5 | 73.49±0.72 | 79.12±0.00 | 48.81±0.10 | 71.18±0.03 | 82.58±0.41 | 80.53±1.38 | 71.10±0.01 |
| AIMC-CVR | **83.52±1.00** | **91.37±0.00** | **51.88±0.08** | **82.29±0.01** | **84.43±1.21** | **81.27±1.81** | **71.88±0.00** |
| Fscore(%) | | | | | | | |
| AIMC-CVR-v1 | 55.37±1.01 | 67.17±0.00 | 34.61±0.09 | 42.93±0.05 | 74.79±1.02 | 54.01±1.49 | 48.46±0.01 |
| AIMC-CVR-v2 | 58.25±0.90 | 66.74±0.00 | 35.16±0.08 | 55.37±0.03 | 73.17±0.58 | 69.53±1.81 | 57.90±0.01 |
| AIMC-CVR-v3 | 31.02±1.31 | 83.99±0.01 | 17.58±0.12 | 18.18±0.17 | 57.54±0.68 | 55.25±1.70 | 50.15±0.00 |
| AIMC-CVR-v4 | 51.70±1.60 | 79.80±0.00 | 35.43±0.12 | 17.94±0.15 | 59.96±0.46 | 63.34±1.43 | 46.26±0.08 |
| AIMC-CVR-v5 | 58.16±0.87 | 66.72±0.00 | 35.06±0.08 | 55.39±0.03 | 73.27±0.50 | 69.67±1.73 | 57.90±0.01 |
| AIMC-CVR | **70.77±1.22** | **87.76±0.00** | **37.05±0.08** | **69.19±0.02** | **77.82±1.03** | **70.97±2.35** | **58.86±0.00** |

