# OpenReview forum: "Alleviate Anchor-Shift: Explore Blind Spots with Cross-View Reconstruction for Incomplete Multi-View Clustering"
_NeurIPS.cc/2024/Conference — NeurIPS 2024 poster_

### Official Review · Reviewer_mjvb · 2024-07-06

**Soundness:** 3
**Presentation:** 3
**Contribution:** 4
**Rating:** 5
**Confidence:** 5

**Summary:**

The paper proposes a cross-view reconstruction-based multi-view clustering algorithm to address the issue of anchor shift in missing data scenarios. Specifically, The method guides anchor learning by reconstructing the missing parts of the data. It uses an affine combination-based reconstruction strategy, rather than a convex combination, to avoid the negative impact of blind reconstruction areas. Theoretical analysis demonstrates the superiority of affine combination constraints over convex combination for this problem. Comparative experiments on multiple datasets with varying missing rates show the effective clustering capability of the proposed algorithm. The key innovations are the cross-view reconstruction approach, the use of affine combination constraints, and the theoretical justification for this choice. The experimental results validate the effectiveness of the proposed method for handling missing data in multi-view clustering tasks.

**Strengths:**

The anchor shift problem studied in this paper is a challenging issue that has been rarely addressed in the multi-view clustering field.
The proposed method effectively addresses the observed problems through the collaboration of two modules, yielding good experimental results.
The theoretical analysis provided in the paper intuitively demonstrates the superiority of the affine combination-based framework over previous convex combination approaches.

**Weaknesses:**

However, the paper contains some inaccurate descriptions, such as stating that matrix decomposition has n^2 complexity in the introduction, which is not entirely accurate.
The paper includes many symbols, but the lack of a symbol table reduces readability.
There is a contradiction regarding the final step of Algorithm 1, where it states k-means is directly applied to Z, while Section 3.2 describes applying k-means to the left singular vector matrix of Z.

**Questions:**

Regarding Algorithm 1, the paper does not discuss the possibility of using spectral clustering instead of k-means in the final step. It's unclear how this substitution would affect the performance of the proposed algorithm.

Additionally, the paper does not clearly explain the relationship between the cross-view learning approach used in this work and the broader multi-view learning field. The connection between these concepts is not well-articulated.

**Limitations:**

See Weaknesses.

---

> ### Author Rebuttal · Authors · 2024-08-06
>
> **1. Inaccurate statements**
>
> Thanks for the comment. Due to the additional introduction of regularization constraints on the entire graph, such as manifold constraints, most existing matrix decomposition methods exhibit an $O(n^2)$ complexity [1][2]. We will revise this statement in the final version.
>
> **2. Notation table**
>
> Thanks for the comment. We have added a symbol table to explain the key symbols used in our paper in the global rebuttal file. This symbol table will also be added in the final version to enhance readability.
>
> **3. Contradiction in Algorithm 1**
>
> Sorry for our mistake. In our method, the final clustering results are produced by applying $k$-means to the left singular vector matrix of $\textbf{Z}$. We will correct it in our final version.
>
> **4. Final process of clustering**
>
> Thanks for the comment. The anchor graph $\textbf{Z}$ obtained from our method is $n \times m$, which cannot be directly used for spectral clustering. According to [3], applying $k$-means to the left singular vectors of $\textbf{Z}$ is equivalent to performing spectral clustering on the doubly stochastic matrix constructed from $\textbf{Z}$. Our method differs from [3] in the constraints applied to $\textbf{Z}$. Therefore, in Section A.4, we prove that the anchor graph $\textbf{Z}$ obtained from our method can also be recovered as a doubly stochastic matrix.
>
> **5. Relationship of cross-view learning and multi-view learning**
>
> Thanks for the comment. Cross-view learning is a subset of multi-view learning. Unlike most previous multi-view learning methods, which build relationships between samples within the single view, cross-view learning aims to explore the relationships between samples across different views. In multi-view alignment methods, cross-view learning is a key technique for aligning samples between different views. In our approach, based on the assumption that samples will always appear in at least one view under missing data scenarios, introducing cross-view learning helps mitigate the impact of missing samples within a single view on anchor point learning.
>
> **References**
>
> [1] Rai N, Negi S, Chaudhury S, et al. Partial multi-view clustering using graph regularized NMF. 2016 23rd International Conference on Pattern Recognition (ICPR). IEEE, 2016: 2192-2197.
>
> [2] Zhao H, Ding Z, Fu Y. Multi-view clustering via deep matrix factorization. Proceedings of the AAAI conference on artificial intelligence. 2017, 31(1).
>
> [3] Wang S, Liu X, Liu L, et al. Highly-efficient incomplete large-scale multi-view clustering with consensus bipartite graph. Proceedings of the IEEE/CVF conference on computer vision and pattern recognition. 2022: 9776-9785.

---

### Official Review · Reviewer_4LaZ · 2024-07-09

**Soundness:** 3
**Presentation:** 3
**Contribution:** 3
**Rating:** 6
**Confidence:** 5

**Summary:**

By employing cross-view anchor learning and affine combination-based reconstruction，the authors propose an incomplete multi-view clustering method to alleviate the anchor-shift problem. Besides, the authors theoretically analysis the advantages of affine combination-based reconstruction, which help to explore blind spots in sample reconstruction. Experimental results on several datasets validate the effectiveness of the proposed method.

**Strengths:**

1.The affine combination-based reconstruction module is a simple yet effective novel approach.

2.The authors provide thorough theoretical and experimental validation of the proposed module's effectiveness.

**Weaknesses:**

1.The authors fail to provide precise definitions for some symbols used in the paper, such as n_p.

2.Some inconsistencies are present in the experimental figures. For example, the x-axes of the first three subfigures in Fig. 3 are spaced at intervals of 20, whereas Fig. 3d uses intervals of 50. Additionally, Fig. 4c and 4d need to be rotated for better presentation.

3.The writing quality needs improvement. The conclusion section is too brief and does not adequately summarize the paper.

**Questions:**

1.Can the authors provide a clearer explanation of Fig. 1, especially in its caption?

2.Why do the last three datasets in Table 4 have identical dimensions? Is this an oversight by the authors?

**Limitations:**

The authors have acknowledged some limitations of their work but have not discussed future work directions.

---

> ### Author Rebuttal · Authors · 2024-08-06
>
> **1. Definition of symbols**
>
> Thanks for the comment. $n_p$ represents the number of samples in the $p$-th view. In the global rebuttal file, we have provided a notation table that explains the main symbols used in the paper. Please refer to the Table 2 in the global rebuttal file.
>
> **2. Inconsistencies of figures**
>
> Thanks for the comment. We have made corrections to the inconsistencies in Fig. 3 and the obscured parts in Fig. 4. Please refer to Figure 1 and Figure 2 in the global rebuttal file.
>
> **3. Rewrite conclusion**
>
> Thanks for the comment. We have rewritten the conclusion as follows to better summarize: In this paper, we introduce an AIMC-CVR method to alleviate the anchor-shift problem in anchor-based incomplete multi-view clustering. Specifically, we introduce a cross-view learning module and a reconstruction module to fix the influence of incomplete data. Additionally, we explore the blind spots in sample reconstruction with affine combination. Experiments and theoretical analysis validate the effectiveness of the proposed AIMC-CVR method. We will replace the conclusion in our final version.
>
>
> **4. Clearer explanation of Fig. 1**
>
> We further explain Fig.1 as follows: (a)Anchors learned in complete data: The true anchors are generated with k-means performed in the complete data. (b)Anchors initialized in incomplete data: The initial anchors are generated with k-means performed in the incomplete data, which is shift the origin point compared with the true anchors. (c)Data reconstructed with convex combination: The convex combination-based reconstructed data is restricted in the convex hull of anchors. (d)Data reconstructed with affine combination: The affine combination-based reconstructed data is breaking through the limitations of convex hull, which can represent any position in affine space.
>
> **5. Explanation of the dataset**
>
> All three datasets consist of image data. We used four operators to extract features for each image, namely LBP (Local Binary Pattern), HOG (Histogram of Oriented Gradient), Gist, and Gabor. These operators provide four different views, each with the same dimensionality on different datasets.

---

### Official Review · Reviewer_fq57 · 2024-07-12

**Soundness:** 3
**Presentation:** 3
**Contribution:** 3
**Rating:** 7
**Confidence:** 4

**Summary:**

This paper proposes an anchor-based incomplete multi-view clustering with cross-view reconstruction (AIMC-CVR). To tackle the anchor-shift induced by incomplete multi-view data, AIMC-CVR reconstructs missing samples with learned anchors. The traditional convex combination is replaced with affine combination for more reconstruction accuracy. Two theorems is proposed to demonstrate the advantages of affine combination.

**Strengths:**

The motivation of this paper is clear, focusing on the anchor shift problem in missing data scenarios, and the corresponding solution is theoretically and experimentally validated.

**Weaknesses:**

The paper emphasizes the missing rates in the experimental data but lacks a detailed description of how the missing data is constructed.

The descriptions of the five variants in the ablation study are not clear enough. Understanding these variants is crucial for validating the effectiveness of the proposed algorithm.

There are some symbol errors in the paper, such as "KTT conditions" which should be "KKT conditions".

**Questions:**

Why use v*v projection matrices? In fact, v matrices projecting data from each view to the same dimension should be sufficient to solve the dimensional inconsistency problem. Could the authors provide further explanation?

In 2021, Yin et al. studied the reconstruction of missing datap[1]. How does the reconstruction method in this paper compare to theirs in terms of advantages?
[1] Yin J, Sun S. Incomplete multi-view clustering with reconstructed views. IEEE TKDE, 2021.

**Limitations:**

The authors have acknowledged some limitations of their work in the appendix and suggested potential solutions.

---

> ### Author Rebuttal · Authors · 2024-08-06
>
> **1. Incomplete data construction**
>
> Thanks for the comment. For the datasets mentioned in our method, we remove some instances on each view randomly to get their incomplete versions. Specifically, with the principle that each instance is present in at least one view, we generate missing datasets at missing rates in intervals of 0.1 from 0.1 to 0.9.
>
> **2. Descriptions of the five variants**
>
> Thanks for the comment. The five variants of our proposed method are constructed as follows: (1) AIMC-CVR-v1 removes the cross-view anchor learning module by setting $\lambda = 0$. (2) AIMC-CVR-v2 removes the affine combination-based reconstruction module. (3) AIMC-CVR-v3 removes the sparsity regularization term by setting $\beta = 0$. (4) AIMC-CVR-v4 keeps the initialized $\mathbf{A}^{(p)}$ fixed and does not update it during subsequent optimization. (5) AIMC-CVR-v5 replaces affine combination with convex combination by adding non-negative constraints to  $\mathbf{Z}^{(p)}$.
>
> **3. Symbol errors**
>
> Sorry for our mistake. We will correct the symbol errors in our final version.
>
> **4. Different projection strategy**
>
> In fact, we only used $v(v-1)/2$ projection matrices because we project lower-dimensional data to higher-dimensional spaces when one view's data dimension is smaller than another's. Unlike previous methods that use $v$ projection matrices to project data into the same lower-dimensional space, our approach aims to preserve high-dimensional information and reduce information loss. To validate the effectiveness of our method, we replace the projection matrix $\mathbf{W}^{(pq)}\in \mathbb{R}^{d_p \times d_q}$ between the $p$-th and $q$-th view with $\mathbf{W}^{(p)}\in \mathbb{R}^{k \times d_p}$ to get the variant AIMC-CVR-V6. The clustering performance of our proposed AIMC-CVR and its variant AIMC-CVR-V6 is shown in Table 1 on the global rebuttal file, which demonstrates the superiority of our projection strategy.
>
> **5. Novelty of imputation strategy**
>
> The imputation method proposed by Yin et al. is constructed based on an $n \times n$ full graph, which incurs $O(n^2)$ space complexity and $O(n^2 log(n))$ time complexity. In contrast, our method achieves imputation based on the anchor graph, with both time and space complexity being linear with respect to $n$, offering a significant efficiency advantage, as demonstrated in Section A.6. Additionally, we expand the reconstruction space of the samples into the affine space of the anchors, which provides a larger search space compared to Yin's method.

---

### Official Review · Reviewer_nWqT · 2024-07-12

**Soundness:** 3
**Presentation:** 2
**Contribution:** 3
**Rating:** 6
**Confidence:** 5

**Summary:**

This paper proposes a novel anchor-based IMVC method called AIMC-CVR to address the anchor-shift caused by missing data. AIMC-CVR consists of two modules: cross-view anchor learning and affine combination-based reconstruction. The former helps in learning a complete anchor graph, while the latter aims to recover the missing data. Carefully designed experiments validate the effectiveness of AIMC-CVR in clustering tasks.

**Strengths:**

-The concept of anchor shift is novel and introduced for the first time in this paper.
-Experiments demonstrate the effectiveness of the proposed method in clustering performance.

**Weaknesses:**

-The paper introduces two modules: cross-view anchor learning and affine combination-based reconstruction. However, it lacks a description of the relationship between these two modules. Are both modules necessary in the proposed method?
-The description of the experimental setup is unclear, particularly the settings in Figure 1. Clear descriptions of the data sources and the anchor generation method are crucial to support the motivation of this paper.

**Questions:**

-What do the gray areas in Figure 1(c) represent? There seems to be a deviation between the reconstructed points in Figure 1(d) and the real points in Figure 1(a). Does this deviation affect subsequent results?
-Why do different datasets show different convergence states in Figure 3? I observed that some converge earlier while others converge later. Can you provide further explanation?

---

> ### Author Rebuttal · Authors · 2024-08-06
>
> **1. The necessity of the proposed modules:**
>
> Thanks for the comment. In AIMC-CVR, both modules are essential for alleviating the anchor-shift problem caused by incomplete data. The cross-view anchor learning module mitigates such problem by leveraging available data across views to learn complete anchor graphs and more accurate anchors. Meanwhile, the affine combination-based reconstruction module focuses on reconstructing missing samples based on the anchors, thus preventing the anchors from being affected by the missing data. These modules cooperate towards the same goal: the former utilizes cross-view complementary information, and the latter employs imputation techniques to alleviate anchor-shift. However, in practice, the reconstruction module depends on the initial anchors learned by the cross-view anchor learning module. Ablation experiments show significant performance drops in versions V1 and V2, where each of these modules is removed, underscoring their necessity.
>
> **2. Experimental setup in Fig.1:**
>
> Thanks for the comment. When generating data, we initially uniformly select four points on a circle with a radius of 1 centered at the origin in a two-dimensional coordinate system as cluster centers. Subsequently, we randomly generate 40 two-dimensional vectors following the standard normal distribution. Each group of 10 vectors is added to the corresponding cluster center, resulting in four classes with a total of 40 data points. Notably, the two-dimensional vectors generated on the two views are distinct but equally important. In Fig. 1a, the true anchors are obtained by applying k-means on the complete data. Fig. 1b shows the initial anchors derived from k-means on the incomplete data. Fig. 1c presents the anchors based on AIMC-CVR-V5, where the affine constraint is replaced with a convex constraint. Fig. 1d displays the anchors learned using the proposed AIMC-CVR method. We will add the above setting in the final version.
>
> **3. Further description of Fig.1:**
>
> Thanks for the comment. In Fig. 1c, the white area represents the convex hull formed by four anchors, indicating that all points within this area can be expressed as a convex combination of the anchors. The gray area represents the blind spots where points cannot be expressed by the convex combination of these anchors. The points in Fig. 1a are the original sample points, whereas the black points in Fig. 1d are reconstructed with our proposed method. Although there are some differences in representation, the reconstructed points in Fig. 1d are evidently closer to the true points compared to those restricted within the convex hull in Fig. 1c. Our objective is not to precisely recover the missing samples, but to mitigate the impact of missing samples on the anchors. Compared to the anchors generated in Fig. 1b, the anchors in Fig. 1d are closer to the true anchors. Thus, while our method can not completely recover the missing samples, it effectively alleviates the anchor-shift problem in incomplete multi-view clustering, achieving state-of-the-art clustering performance.
>
> **4. Difference in convergence states on different datasets:**
>
> Thanks for the comment. We speculate that the differences may arise from the following three factors: Firstly, different types of data naturally have inherent differences, leading to varying numbers of iterations required to meet the convergence criteria during optimization. Secondly, when fusing multi-view data, the similarity between views varies across datasets. Data with more similar views converge faster during integration, while data with greater view differences take longer. Thirdly, the alternating optimization algorithm is highly influenced by the initial value settings, with different initial settings resulting in different numbers of iterations.

---

> > ### Comment · Reviewer_nWqT · 2024-08-13
> >
> > Thanks for your responses. Most of my concerns have been addressed. I will raise my score.

---

### Author Rebuttal · Authors · 2024-08-06

We thank the SAC, AC, and PCs for their efforts and constructive comments, which are helpful in further improving the quality of our manuscript. We respond to your questions carefully one by one carefully, and we hope our responses can address your concerns.

Note that there are two tables and two figure in the attached PDF, corresponding to RQ4 for Reviewer fq57, RQ1 and RQ2 for Reviewer 4LaZ, and RQ2 for Reviewer mjvb.

---

### Author Response · Authors · 2024-08-11
**Paper Discussion**

Dear Reviewers, Area Chairs, Senior Area Chairs and Program Chairs,

We sincerely thank the efforts and constructive comments you have made for this paper. The reviewers put forward many insightful questions and valuable suggestions towards improving our paper.

In the rebuttal phase, we provided detailed responses to all reviewers' comments point by point, hoping to address the issues raised by reviewers nWqT, fq57, 4LaZ and mjvb, including supplementing experimental details, explaining mainly used notations, further refining and describing charts, and adding a comparative experiment to validate the effectiveness of the proposed projection strategy.

The discussion period is coming to an end, and we are actively waiting for further discussion from Reviewers. If you have any questions, we are happy to discuss them with you at any time.

Thanks & Regards,

Authors of paper-3388.

---

### Decision · Program_Chairs · 2024-09-25

**Decision:**

Accept (poster)

**Comment:**

This paper proposes a novel anchor-based method for incomplete multi-view clustering. Specifically, the authors propose a cross-view reconstruction strategy to recover missing samples. The reviewers believe that the proposed method is simple yet efficient, and empirical studies demonstrate its superiority.  The reviewers have reached an agreement to accept this paper. Hence, it should be a clear acceptance. The authors are required to follow the reviewers' suggestions to polish the main paper.